Not all jellyfish are equal: isotopic evidence for inter- and intraspecific variation in jellyfish trophic ecology

Fleming Nicholas E.C. 1 2
Harrod Chris 1 3 chris@harrodlab.net
Newton Jason 4
Houghton Jonathan D.R. 1 2 5
1 School of Biological Sciences, Medical Biology Centre, Queen’s University Belfast , Belfast , UK
2 Queen’s University Belfast Marine Laboratory , Portaferry, Co. Down , UK
3 Fish and Stable Isotope Ecology Laboratory, Instituto de Ciencias Naturales Alexander von Humboldt, Universidad de Antofagasta , Antofagasta , Chile
4 NERC Life Sciences Mass Spectrometry Facility, Scottish Universities Environmental Research Centre , East Kilbride , UK
5 Institute for Global Food Security, Queen’s University Belfast , Belfast , UK
Stanford Jack
Electronic publication date: 2015 Jul 21
Publication date: 2015
Volume: 3
Electronic Location ID: e1110
Received 2015 Feb 26; Accepted 2015 Jun 27
Copyright: © 2015 Fleming et al.
Copyright year: 2015
Copyright holder: Fleming et al.
License: This is an open access article distributed under the terms of the Creative Commons Attribution License, which permits unrestricted use, distribution, reproduction and adaptation in any medium and for any purpose provided that it is properly attributed. For attribution, the original author(s), title, publication source (PeerJ) and either DOI or URL of the article must be cited.
License URL: https://creativecommons.org/licenses/by/4.0/

Keywords: Scyphozoan jellyfish, Food web, Aurelia aurita, Cyanea lamarckii, Cyanea capillata, Niche width, Bayesian statistics

Funding: Northern Ireland Assembly NERC This study was supported by a Department of Education and Learning (DEL) PhD studentship awarded to NECF by the Northern Ireland Assembly. Access to the Life Sciences Mass Spectrometry Facility was supported by NERC. The funders had no role in study design, data collection and analysis, decision to publish, or preparation of the manuscript.

==============================
Jellyfish are highly topical within studies of pelagic food-webs and there is a growing realisation that their role is more complex than once thought. Efforts being made to include jellyfish within fisheries and ecosystem models are an important step forward, but our present understanding of their underlying trophic ecology can lead to their oversimplification in these models. Gelatinous zooplankton represent a polyphyletic assemblage spanning >2,000 species that inhabit coastal seas to the deep-ocean and employ a wide variety of foraging strategies. Despite this diversity, many contemporary modelling approaches include jellyfish as a single functional group feeding at one or two trophic levels at most. Recent reviews have drawn attention to this issue and highlighted the need for improved communication between biologists and theoreticians if this problem is to be overcome. We used stable isotopes to investigate the trophic ecology of three co-occurring scyphozoan jellyfish species (Aurelia aurita, Cyanea lamarckii and C. capillata) within a temperate, coastal food-web in the NE Atlantic. Using information on individual size, time of year and δ13C and δ15N stable isotope values, we examined: (1) whether all jellyfish could be considered as a single functional group, or showed distinct inter-specific differences in trophic ecology; (2) Were size-based shifts in trophic position, found previously in A. aurita, a common trait across species?; (3) When considered collectively, did the trophic position of three sympatric species remain constant over time? Differences in δ15N (trophic position) were evident between all three species, with size-based and temporal shifts in δ15N apparent in A. aurita and C. capillata. The isotopic niche width for all species combined increased throughout the season, reflecting temporal shifts in trophic position and seasonal succession in these gelatinous species. Taken together, these findings support previous assertions that jellyfish require more robust inclusion in marine fisheries or ecosystem models.

Introduction

Jellyfish (here considered as Phylum Cnidaria; Class Scyphozoa) are a conspicuous, yet long-overlooked component of pelagic marine systems. In recent years, the notion of gelatinous species as merely carbon sinks or trophic dead ends has become largely obsolete (Arai, 2005; Hansson & Norrman, 1995), and there is renewed interest in their trophic ecology (Stoner & Layman, 2015; Sweetman et al., 2014). Beyond widely-recognised obligate predators of jellyfish such as leatherback turtles (Houghton et al., 2006), Arai (2005) drew attention to a wide range of opportunistic carnivores such as molluscs, arthropods, reptiles and birds that feed upon gelata episodically. More recently, opportunist scavenging on jellyfish has been observed in the deep-sea (Sweetman et al., 2014) as well as shallower benthic environments (Stoner & Layman, 2015). From a perspective of top-down control, it is also known that the collective prey-consumption rates of gelatinous aggregations can be so high that predation can directly or indirectly control the population size of other zooplanktonic organisms including larval fish (Nielsen, Pedersen & Riisgård, 1997; Purcell, 1992). Moreover, evidence of sized-based trophic shifts in the moon jellyfish Aurelia aurita (Linnaeus, 1758) (Fleming et al., 2011; Graham & Kroutil, 2001) suggest that jellyfish could themselves exhibit size-associated shifts in trophic ecology, e.g., similar to those shown by fishes (Graham et al., 2007).

Prompted by a growing body of evidence, Pauly et al. (2009) stressed that the functional role of gelatinous taxa requires more robust inclusion in marine fisheries or ecosystem models. At present, such species are typically considered as a single functional group or an ‘average’ group of animals, feeding on the same prey throughout their life history (Boero et al., 2008; Pauly et al., 2009). Indeed, out of 100 models considered, only 23% incorporated jellyfish as a distinct functional group (normally feeding at a single trophic level) and only 4% of models considered them in any greater detail, e.g., feeding at two trophic levels (Pauly et al., 2009). Consequently, seasonal or ontogenetic shifts in diet (Fleming et al., 2011; Graham & Kroutil, 2001), intra-specific differences in prey types (Fancett, 1988) and intra-guild predation (Bayha et al., 2012; Robison, 2004; Titelman et al., 2007) are typically over-simplified or disregarded entirely. Pauly et al. (2009) and Doyle et al. (2014) have made a number of suggestions for researchers working with gelatinous species on how to generate data that are useful to theoreticians. These studies highlight the fact that the ecological-modelling community cannot be expected to consider jellyfish in adequate detail, if the data required are not provided by other researchers (Doyle et al., 2014). This is a valid point, but until recently many questions surrounding the trophodynamics of gelatinous species appeared intractable, given the spatial and temporal variability of aggregations (Doyle et al., 2007; Houghton et al., 2007), the broad-scale over which they can occur (Doyle et al., 2008) and methodological limitations (Purcell, 2009).

Within this broad context, the aim of this study was to examine trophic variation in three sympatric jellyfish species (Aurelia aurita (Linnaeus, 1758), Cyanea lamarckii (Péron & Lesueur, 1810) and C. capillata (Linnaeus, 1758)) in a temperate coastal marine system. Strangford Lough in Northern Ireland was identified as an ideal study system as it supports an annual succession of gelatinous zooplankton species from early May to late August (Fleming, Harrod & Houghton, 2013). We used stable isotopes (δ13C and δ15N) to consider size-based and temporal shifts in the trophic ecology of the three jellyfish species, both individually and collectively as a dominant large gelatinous zooplankton community. Isotopic approaches have been used widely to examine the trophic ecology of marine and estuarine systems in general (Peterson & Fry, 1987), and are gathering momentum for the study of gelatinous species (Kogovšek et al., 2014; Nagata et al., 2015; Pitt et al., 2008). To provide data that might aid the further inclusion of jellyfish into ecosystem models, our analyses were aligned to examine three specific questions: (1) could all jellyfish be considered as a single functional group or was there evidence for distinct inter-specific differences in trophic ecology?; (2) were size-based shifts in trophic ecology found previously in A. aurita a common trait across species?; and (3) when considered collectively, did the trophic position and isotopic niche of three sympatric species remain constant over time?

Materials & Methods

Collection and processing

Strangford Lough (54°28′20.98″N 5°35′10.60″W; Northern Ireland) is a large, semi-enclosed coastal embayment (150 km2) that flows into the northern Irish Sea (see Maloy et al., 2013 for a description). Three scyphozoan jellyfish species are persistently present in the lough but their relative abundance varies over time. In May, the community is typically dominated by Aurelia aurita, with an increase in Cyanea lamarckii in early June and Cyanea capillata in July (Fleming et al., 2014). All three species disappear from the water column in the same order from late July onwards (Fleming et al., 2014; Fleming, Harrod & Houghton, 2013). Medusae of these three jellyfish species were sampled monthly from Strangford Lough (May 2010 to September 2010). Jellyfish were collected near the surface from a small boat using a dip net (mesh size 1 mm) for smaller jellyfish and a larger net (5 mm mesh size) for larger individuals. Sampling was conducted in a non-random manner, as our aim was to collect sufficient individuals to allow for balanced statistical comparisons (e.g., across months). Unfortunately, owing to temporal variation in the abundance of the different species, and often challenging weather conditions, it was not possible to ensure a balanced number of samples per species.

Filter-feeding bivalves (Mytilus spp.) and grazing gastropods (Littorina saxatilis (Olivi)) were sampled over the study period from intertidal areas adjacent to the jellyfish sampling sites over the same period (Woodland et al., 2012). These species are long-lived, dominant and ubiquitous, providing a measure of isotopic baselines of the pelagic (bivalve) and benthic (gastropods) primary production pathways as suggested by Post (2002) and supported by others (e.g., Mallela & Harrod, 2008; Richoux & Ndhlovu, 2014). Furthermore, isotopic turnover rates (expressed as half-life) in the moon jellyfish (Aurelia aurita) recently described by D’Ambra, Carmichael & Graham (2014) who estimated a half-life for δ13C (10.8 days) and δ15N (9.7 days) are similar to that of Mytilus (δ13C = 9 days; δ15N = 14 days) (Dubois et al., 2007), suggesting a similar ability to track temporal shifts in baseline isotope values.

Laboratory and SIA analysis

All jellyfish samples were collected and processed immediately to prevent potentially marked effects of freezing and ethanol preservation (Fleming et al., 2011). A. aurita, C. lamarckii and C. capillata were weighed and measured (wet mass: ±1 g; bell diameter: ±1 cm), then medusae were rinsed thoroughly in filtered seawater, after which bell (mesoglea) tissues were separated and dried at 60 °C in a drying oven following Fleming et al. (2011). Samples were ground to a fine powder in an agate mortar and pestle and then weighed into tin cups prior to stable isotope analysis. Preliminary analyses revealed that optimal sample mass for mass spectrometry varied between taxa i.e., A. aurita ≈ 12 mg; C. lamarckii ≈ 2.4 mg, C. capillata ≈ 5.1 mg and other taxa ≈ 0.8 mg). Samples were analysed for δ13C, δ15N and C:N at the East Kilbride Node of the Natural Environment Research Council Life Sciences Mass Spectrometry Facility via continuous flow isotope ratio mass spectrometry using an ECS 4010 elemental analyser (Costech, Milan, Italy) interfaced with a Delta XP mass spectrometer (Thermo Electron, Bremen, Germany). The standard deviation of multiple analyses of an internal gelatine standard was ∼0.1‰ for both δ13C and δ15N.

Statistical analysis

Prior to analysis, bell mass, bell diameter and stable isotope data were log10-transformed to improve normality and reduce heteroscedasticity (δ13C data were log10 + 40 transformed due to their negative values). Recently evidence has emerged that air-drying gelatinous tissue can result in 15N enrichment in more proteinaceous species (Kogovšek et al., 2014). C:N ratios of the three species were compared and found not to differ (F2,120 = 1.48, P = 0.232), suggesting that any effect of air-drying would be consistent across species. We used various statistical approaches to characterise and compare the trophic ecology (inter-specific, intra-specific and community) of the jellyfish species.

Permutational multivariate analysis of variance (PERMANOVA) (Anderson, 2001; Anderson, Gorley & Clarke, 2008) in PRIMER 6.1.12 (Clarke & Gorley, 2006; Clarke & Warwick, 2001) was used to examine variation in the location of centroids of log−10-transformed δ15N–δ13C data, based on a Euclidean similarity matrix (npermutations = 9,999). PERMANOVA was used to examine variation in bell δ15N and δ13C values by species (inter-specific variation) and sample month (intra-specific variation). Here, it is assumed that where δ15N–δ13C centroids overlap (i.e., are not significantly different), then trophic ecology is similar e.g., between species or month. As some small (n ≤ 3) sample sizes were recorded for species across the different months (C. capillata in May; A. aurita and C. lamarckii in August), it was not possible to make a balanced two-way analysis for the entire study period. A full two-way PERMANOVA examining isotopic variation associated with Species and Month (and the Species × Month interaction) was conducted for June and July only. One-way PERMANOVA was used to compare variation within species across months.

Two-way PERMANOVA was used to examine whether δ15N–δ13C values from baseline indicators associated with the pelagic and benthic pathways varied either between functional groups or over time (survey month). We also conducted a similar univariate two-way PERMANOVA comparing temporal shifts in δ15N data from the two functional groups in order to examine whether shifts in jellyfish δ15N were related to changes at the base of the food web or in apparent jellyfish trophic level.

As jellyfish are often considered as a single functional group, we examined how an indicator of community level trophic position varied across the survey period by pooling δ15N data from all three jellyfish species and conducting a univariate PERMANOVA with month as a fixed independent factor.

We used the SIBER procedure (Stable Isotope Bayesian Ellipses in R) (Jackson et al., 2011) within the R package SIAR (Parnell et al., 2010) to examine variation in jellyfish isotopic niche space. This approach relies on the concept that multiple stable isotope ratios measured from consumers represent niche dimensions, e.g., variation in δ13C reflects use of different energy sources, or habitats, while δ15N provides information on the trophic level at which a consumer feeds (Peterson & Fry, 1987). This so called ‘isotopic niche’ or ‘δ-space’ (Newsome et al., 2007) is thought to reflect the trophic niche of groups of consumers (Bearhop et al., 2004; Fink et al., 2012; Layman et al., 2007), where more isotopic variation reflects a larger consumer isotopic niche, assuming that spatial or temporal variation in baseline isotopic values is considered. Here we use Bayesian Standard Area Ellipses (SEAB), as the use of Bayesian inference allows the incorporation of uncertainty such as small sample sizes (Jackson et al., 2011). This iterative approach uses Monte Carlo Markov-Chain simulation to construct ellipses characterising isotopic variation that provide a robust indicator of isotopic niche width. We used this technique to characterise temporal variation in the trophic niche of the three jellyfish species, as well as overlap between species. We also examined temporal variation in SEAB values calculated for the jellyfish community as a whole (i.e., all three species of jellyfish combined). In order to examine the differences in isotopic niche area (SEAB) between different consumer groups, we calculated probabilities from posterior distributions (based on 100,000 draws) of the parameters of model M given the prior data D (Pr(M|D)). These maximum likelihood comparisons provide direct probabilities of differences rather than the traditional frequentist test of a null-hypothesis. In order to differentiate these comparisons, maximum-likelihood based probabilities are reported here as percentages.

In stable isotope studies, consumer trophic position is typically estimated from δ15N data, which are corrected for baseline variation and trophic fractionation (Post, 2002). Although we had reliable data on pelagic and benthic δ15N baselines (see above), information on jellyfish trophic enrichment factors (TEFs) is extremely limited. D’Ambra, Carmichael & Graham (2014) recently provided TEFS for A. aurita, in what represents the only experimental estimate of jellyfish trophic fractionation in the literature. The mean ± SD TEFs estimated by D’Ambra et al. for A. aurita (Δ13C = 4.3 ± 0.2‰; Δ15N = 0.1 ± 0.2‰) are very unusual and contrast markedly with the average TEFS more commonly seen in the literature (e.g., Post (2002): Δ13C = 0.4 ± 1.3‰; Δ15N = 3.4 ± 1‰; McCutchan et al. (2003) (Δ13C = 0.5 ± 1.3‰, Δ15N = 2.3 ± 1.5‰)). As use of the jellyfish specific TEFs provided by D’Ambra, Carmichael & Graham (2014) resulted in unfeasibly high trophic positions for the jellyfish species, including A. aurita, we did not make direct estimates of trophic position, but provide indirect estimates by presenting δ15N data.

Finally normal linear least-squares regression was used to examine how log10 transformed stable isotope values (δ13C data were log10 + 40 transformed) varied with individual size (bell wet mass and diameter).

Statistical analyses were conducted using routines in PRIMER-E 6 (Clarke & Gorley, 2006) and SYSTAT 13.1 (SYSTAT Software Inc, 2009). SIBER analyses (Jackson et al., 2011) were conducted using SIAR (Parnell et al., 2010) in R version 3.1.2 (R Development Core Team, 2014). An alpha level of 0.05 is used throughout to indicate statistical significance.

Results

Baseline variation

Comparisons of baseline indicator (filter feeding and grazing molluscs) δ15N–δ13C values across the study period using two-way PERMANOVA showed strong evidence of isotopic differences between the two functional groups (Pseudo-F1,108 = 82.44, P = 0.0001), but less evidence for marked temporal differences (Month: Pseudo-F2,108 = 2.64, P = 0.06). There was no evidence for a significant interaction between these two factors (Pseudo-F2,108 = 0.04, P = 0.99), indicating that the isotopic differences between the two functional groups were maintained over time.

We also examined δ15N values from filter feeding and grazing molluscs as they provide a reference for measurements of consumer trophic position relative to the base of the food web. Baseline δ15N values differed between the two functional groups (Pseudo-F1,108 = 59.57, P = 0.0001) with benthic grazers (mean ± SD δ15N = 11.2 ± 1.08, n = 58) being 15N enriched by 1.5‰ relative to filter feeding bivalves (bivalve = 9.7 ± 0.7, n = 56) but were consistent across the study period (PERMANOVA on log10-transformed δ15N data; Month: Pseudo-F2,108 = 0.48, P = 0.725). The lack of an interaction between the two factors (Month × Functional Group: Pseudo-F2,108 = 0.087, P = 0.91) indicated that the differences in δ15N between the two functional groups remained constant over time.

Inter-specific variation

A total of 122 medusae were collected from the surface of the water column comprising Aurelia aurita (n = 43), Cyanea lamarckii (n = 36) and C. capillata (n = 43). Data collected across the entire study for the three jellyfish species (Fig. 1) showed considerable intraspecific variation and apparent isotopic overlap between the species. However, when δ15N and δ13C data for individual species were compared over time, differences became apparent (Table 1; Fig. 2).

Figure 1 Isotopic variation in 3 species of co-occuring jellyfish.

Variation in δ13C and δ15N shown in three species of jellyfish over the whole study period. (See Table 1 for summary statistics).

Figure 2 Temporal variation in jellyfish δ13C and δ15N.

Box-whisker plots showing variation in δ13C (A) and δ15N (B) in the three jellyfish species, and within the dominant gelatinous zooplankton community (GZ; all three species combined) over the study period. See Table 1 for sample sizes and other summary statistics. NB: Baseline δ15N values remained constant over this period, indicating that the increase in δ15N values reflected a shift in trophic position rather than seasonal shifts at the base of the food web. Boxes show inter-quartile range, and the bold horizontal bar indicates the median value. Whiskers reflect values 1.5× the interquartile range.

Table 1 Summary statistics for bell stable isotope and C:N ratios.

Species	n	δ13C (±SD)‰	δ15N (±SD)‰	C:N (±SD)	
Aurelia aurita May	16	−20.3 (0.5)	8.5 (1.1)	3.8 (0.1)	
Aurelia aurita June	18	−18.2 (0.5)	10.3 (1.5)	3.5 (0.4)	
Aurelia aurita July	9	−18.1 (0.7)	11.5 (1.5)	3.5 (0.4)	
Aurelia aurita August	2	−17.3 (0.1)	11.8 (1.7)	3.7 (0.1)	
Overall mean A. aurita	43	−19.0 (1.2)	9.7 (1.6)	3.6 (0.2)	
Cyanea lamarckii May	7	−21.4 (0.2)	8.6 (0.6)	3.9 (0.1)	
Cyanea lamarckii June	21	−19.5 (0.7)	11.5 (1.5)	3.7 (0.4)	
Cyanea lamarckii July	5	−19.4 (0.8)	12.1 (1.3)	3.7 (0.3)	
Cyanea lamarckii Aug	3	−19.2 (0.8)	11.5 (0.8)	3.7 (0.2)	
Overall mean C. lamarckii	36	−19.8 (1.0)	11.0 (1.8)	3.7 (0.3)	
Cyanea capillata May	2	−21.4 (0.1)	7.7 (0.1)	3.8 (0.1)	
Cyanea capillata June	13	−19.5 (1.2)	11.0 (2.1)	3.6 (0.4)	
Cyanea capillata July	14	−19.4 (1.1)	12.8 (1.3)	3.6 (0.2)	
Cyanea capillata Aug	16	−18.7 (1.6)	13.3 (1.1)	3.5 (0.3)	
Overall mean C. capillata	43	−19.7 (1.3)	12.4 (1.8)	3.6 (0.1)	

A full two-way PERMANOVA comparing the influence of survey month and species was only possible for all three species in the months of June and July when medusae of all species were present. The analysis of log10-transformed data revealed that δ15N–δ13C centroid location varied significantly between the three jellyfish species (Pseudo-F2,71 = 5.01, P = 0.006) and survey month (Pseudo-F1,71 = 5.1, P = 0.02). However, there was no interaction between species and survey month (F2,71 = 0.25, P = 0.82) indicating that temporal shifts in δ13C–δ15N isotope values were similar across the three scyphozoan species in June and July. Pairwise comparisons showed that A. aurita were isotopically distinct from both Cyanea species in June (C. lamarckii P ≤ 0.0043; C. capillata P = 0.02), and from C. lamarckii in July (P = 0.03). The δ15N–δ13C centroids of the two Cyanea species overlapped during these months (June: P = 0.89; July: P = 0.43).

Next, we considered inter-specific differences in isotopic niche width (Fig. 3). Between-species comparisons (data pooled from all months) showed that C. capillata had the largest mean (95% credibility limits) isotopic niche width of 6.90 (4.95–9.03)‰2, compared to A. aurita (4.94 (3.55–6.46)‰2) or C. lamarckii (5.49 (3.84–7.32)‰2). Maximum-likelihood pairwise comparisons indicated a borderline probability (Probability (P) = 94%) that across the entire study the isotopic niche width of C. capillata was larger than that of A. aurita. There was no statistical support (P = 85%) for differences between C. capillata, and its congeneric C. lamarckii. There was a 67% probability of differences in isotopic niche width size between A. aurita and C. lamarckii.

Figure 3 Variation in isotopic niche width (SEAB) between species (A. a, A. aurita; C. l, C. lamarckii; C. c, C. capillata) and within the dominant gelatinous zooplankton community (GZ; all three species combined) sampled over the survey period.

Boxes represent the 50, 75 and 95% Bayesian credibility intervals estimated from 100,000 draws. Samples marked with * included less than 10 individuals (see Parnell et al., 2010). See Table 3 for statistical comparisons.

Intra-specific variation

Although A. aurita were captured in each of the survey months (Fig. 2), sufficient samples for analysis were not recorded in August (n = 2), and statistical comparisons here are limited to the period May-July (See Table 1 for sample sizes). During this period, the location of A. aurita δ15N–δ13C centroids varied significantly (One-way PERMANOVA Pseudo-F2,38 = 15.19, P = 0.0001), indicating that A. aurita underwent an isotopic shift over the study period. Pairwise tests showed that δ15N–δ13C centroids shifted between May and both June (t = 4.49, P = 0.0002) and July (t = 4.77, P = 0.0001). δ15N–δ13C values overlapped in June and July (t = 1.6, P = 0.12). The difference between May and the other months reflected enrichment in 13C and to a lesser degree 15N from May to the later months.

Sample sizes in C. lamarckii were relatively low throughout the study, with large numbers only being encountered in June (Table 1). C. lamarckii showed significant temporal shifts in the location of the δ15N–δ13C centroids (May–July: Pseudo-F2,31 = 15.46, P = 0.0001). Pairwise tests revealed that centroids differed between May and both June (t = 5.15, P = 0.0002) and July (t = 6.58, P = 0.001), but overlapped between June and July (t = 0.63, P = 0.56). Isotopically, C. lamarckii became increasingly 13C and 15N enriched over the survey period (Fig. 2 and Table 1).

Only two C. capillata were available for analysis in May, but in the following months, δ15N–δ13C centroids for this species changed significantly (June–August: Pseudo-F2,38 = 4.44, P = 0.008). Pairwise tests indicated that this shift was relatively gradual, with isotopic overlap in June and July (t = 1.87, P = 0.06) and July–August (t = 1.22, P = 0.22). Isotopic differences were most marked at the extremes of the collection period: June–August (t = 2.79, P = 0.003).

Bayesian estimates of isotopic niche width (SEAB) showed significant variation within species during the study period (Table 3 and Fig. 3). Pairwise comparisons showed that A. aurita mean isotopic niche width was lower in May relative to other months (Table 3 and Fig. 3), with a 95% probability of a difference from June and a 98% probability of a difference from July. The isotopic niche width of C. lamarckii was reduced in May relative to June (P = 99%) and July (P = 96%), but there were no obvious differences in isotopic niche width in June and July (P = 46%). C. capillata was not recorded in sufficient numbers in May to allow analyses, but showed a similar isotopic niche width through the June–August period (P range 50–60%).

Table 2 Summary statistics for least squares regressions examining relationships between individual jellyfish size and bell stable isotope ratios (mass, length and δ15N data log10 transformed, δ15C data log10 + 40 transformed).

NB: in all cases slopes were significantly different from 1.

Species	Isotope	Comparison	Intercept (±SE)	Slope (±SE)	R 2	F	P	
A. aurita	δ13C
(−21.1 to −17.2‰)	Bell diameter
(6 to 36 cm)	1.224 (0.019)	0.079 (0.015)	0.39	F1,41 = 26.3	<0.001	
A. aurita	δ15N
(6.7 to 14.8‰)	Bell diameter
(6 to 36 cm)	0.609 (0.056)	0.305 (0.045)	0.53	F1,41 = 46.2	<0.001	
A. aurita	δ13C
(−21.1 to −17.2‰)	Wet mass
(12 to 1,702 g)	1.256 (0.013)	0.029 (0.006)	0.40	F1,41 = 26.9	<0.001	
A. aurita	δ15N
(6.7 to 14.8‰)	Wet mass
(12 to 1,702 g)	0.730 (0.038)	0.111 (0.016)	0.54	F1,41 = 48.8	<0.001	
C. lamarckii	δ13C
(−21.6 to −18.5‰)	Bell diameter
(4 to 20 cm)	1.287 (0.019)	0.018 (0.019)	0.02	F1,35 = 0.85	=0.363	
C. lamarckii	δ15N
(7.7 to 15.8‰)	Bell diameter
(4 to 20 cm)	0.939 (0.067)	0.103 (0.066)	0.06	F1,35 = 2.4	=0.131	
C. lamarckii	δ13C
(−21.6 to −18.5‰)	Wet mass
(3 to 493 g)	1.293 (0.013)	0.006 (0.007)	0.02	F1,35 = 0.71	=0.405	
C. lamarckii	δ15N
(7.7 to 15.8‰)	Wet mass
(3 to 493 g)	0.985 (0.047)	0.030 (0.025)	0.04	F1,35 = 1.50	=0.229	
C. capillata	δ13C
(−21.8 to −17.2‰)	Bell diameter
(6 to 85 cm)	1.233 (0.020)	0.062 (0.014)	0.32	F1,41 = 19.1	<0.001	
C. capillata	δ15N
(7.6 to 16.1‰)	Bell diameter
(6 to 85 cm)	0.876 (0.046)	0.157 (0.034)	0.34	F1,41 = 22.0	<0.001	
C. capillata	δ13C
(−21.8 to −17.2‰)	Wet mass
(19 to 23,680 g)	1.259 (0.015)	0.020 (0.005)	0.28	F1,41 = 16.1	<0.001	
C. capillata	δ15N
(7.6 to 16.1‰)	Wet mass
(19 to 23,680 g)	0.931 (0.035)	0.055 (0.012)	0.35	F1,41 = 22.1	<0.001	

Table 3 Bayesian comparisons of isotopic niche width (SEAB) between different jellyfish species and survey months.

Probabilities (based on 100,000 draws) that isotopic niche area in Group A is larger than the comparative value in Group B (A > B) are shown.

Group							Group A				
		A. a May	A. a June	A. a July	C. l May	C. l June	C. l July	C. c June	C. c July	C. c August	
	A. a May	–	0.951	0.980	0.388	0.996	0.969	0.998	0.999	0.999	
	A. a June		–	0.756	0.062	0.855	0.728	0.927	0.938	0.969	
	A. a July a			–	0.029	0.540	0.496	0.697	0.703	0.775	
	C. l May a				–	0.988	0.964	0.993	0.994	0.997	
Group B	C. l June					–	0.460	0.713	0.722	0.821	
	C. l July a						–	0.683	0.688	0.754	
	C. c June							–	0.497	0.596	
	C. c July								–	0.609	
	C. c August									–	
Notes.

Species codes

A. a A. aurita

C. l C. lamarckii

C. c C. capillata

a Groups reflect samples sizes <10.

Both A. aurita and C. capillata showed positive linear relationships between log10-transformed δ13C and wet mass (Table 2, Fig. 4: A. aurita F1,41 = 26.9, R2 = 0.40, P < 0.001; C. capillata F1,41 = 16.1, R2 = 0.28, P < 0.001) and bell diameter (A. aurita F1,41 = 26.3, R2 = 0.39, P < 0.001; C. capillata F1,41 = 19.1, R2 = 0.32, P < 0.001), indicating a shift in dietary source with size in these species. However, there was no evidence for any such relationship in C. lamarckii for wet mass (F1,35 = 0.71, R2 = 0.02, P = 0.405) or bell diameter (F1,35 = 0.85, R2 = 0.02, P = 0.363), indicating that individuals of all sizes assimilated carbon from a similar range of sources. δ15N increased with size (Fig. 4 & Table 2) in both A. aurita (log10-transformed wet mass F1,41 = 48.8, R2 = 0.54, P < 0.001; bell diameter F1,41 = 46.2, R2 = 0.53, P = < 0.001) and C. capillata (wet mass F1,41 = 22.1, R2 = 0.35, P = < 0.001; bell diameter F1,41 = 22.0, R2 = 0.34, P < 0.001). In all cases, the slope of the log10–log10 relationship was <1 (Table 2). As in the case of δ13C, C. lamarckii showed no evidence of any size-based shift in δ15N (wet mass =F1,35 = 1.50, R2 = 0.04, P = 0.229; bell diameter F1,35 = 2.4, R2 = 0.06, P = 0.131).

Figure 4 Figure showing isotopic variation with size.

Variation in bell δ13C (A & B) and δ15N (C & D) with bell diameter (A & C) and wet mass (B & D). Note use of logarithmic scale on x-axes.

Variation at a whole community level

As baseline δ15N values were consistent over time (see ‘Baseline variation’ above), we were able to use δ15N as an indirect indicator of changes in whole community apparent trophic position over time in the absence of reliable TEFs. δ15N values for the dominant gelatinous zooplankton community (All GZ) as measured here, varied over the study period (One-way univariate PERMANOVA Pseudo-F3,119 = 36.9, P = 0.0001; Fig. 2), and showed relative increases in apparent trophic position (δ15N) over time. Pairwise tests showed δ15N in May was lower than in all other months (June, t = 6.2, P = 0.0001; July, t = 10.6, P = 0.0001; August, t = 13.3, P = 0.0001). June δ15N values were higher than May, but lower than subsequent months (May, t = 6.2, P = 0.0001; July, t = 3.1, P = 0.0027; August, t = 4.4, P = 0.002). There was no measurable difference in whole community δ15N values in July and August (t = 1.9, P = 0.07; Fig. 2).

We also examined temporal variation in the community isotopic niche width by pooling values from the three jellyfish species (See all GZ values in Fig. 3). Mean (95% credibility limits) jellyfish isotopic niche width in May was lower than in June, July or August (P = 100% in all cases). However, isotopic niche for the combined jellyfish species began to change in position and width as the season progressed with an increase in isotopic niche (‰2 95% credibility limits) from May = 2.05 (1.31–2.89) to Aug = 5.72 (3.49–8.3), suggesting a broader trophic niche in the latter months (P July >June = 54%; P August >June = 76%; P August >July = 70%).

Discussion

Pauly et al. (2009) described jellyfish as arguably the most important predators in the sea. There is little ambiguity in this statement which, in part, prompted the present study. There is no doubt that the potential expansion of jellyfish in highly depleted oceans is a matter of grave concern (Lynam et al., 2006; Purcell, Uye & Lo, 2007), and an underlying knowledge of how jellyfish function within marine systems is required, so that long-standing trends in populations and communities can be teased apart from shifts in ecosystem structure. Stable isotope analysis offers a powerful biochemical approach to the estimation of trophic and dietary composition of individuals through to communities (Bearhop et al., 2004; Bolnick et al., 2003) and the results presented here support the idea that jellyfish play a more complex trophic role than once envisaged.

Consistency in baseline isotope values

Variation in δ15N–δ13C values measured from baseline indicators of the pelagic (filter feeding bivalve) and benthic (grazing gastropod) energy pathways was driven by functional group rather than survey month. This indicates that any temporal differences observed in jellyfish isotope values and the measures derived from them (i.e., isotopic niche space), reflected changes in jellyfish diet over time rather than shifts at the base of the food web.

Inter-specific differences in trophic ecology

At the whole-study level, isotopic differences were evident between the three jellyfish species in terms of δ15N and δ13C, with post-hoc comparisons highlighting differences between A. aurita and both Cyanea species in June, and with C. lamarckii in July. Conversely, the Cyanea species showed isotopic overlap during June and July. Comparisons of isotopic niche width showed that differences were most marked between A. aurita v C. capillata. Taken together, these results suggest differences in jellyfish behaviour and their capacity to capture and ingest a range of prey items between these two genera (Figs. 2 and 3).

Typically, scyphozoan jellyfish encounter rather than detect and pursue prey and use both ‘passive ambush’ and ‘feeding current’ feeding strategies with direct interception and filtering through tentacles being used in both cases (Kiørboe, 2011). Feeding currents are generated by pulsation of the bell which varies in shape and size between species, with slower velocities normally associated with smaller individuals (Costello & Colin, 1994; Costello & Colin, 1995; Kiørboe, 2011). Depending on the escape velocities of putative prey, differences in feeding current velocity between different jellyfish species might lead to different prey being captured and ingested; however, further work is required to link trophic position with morphological characteristics in an empirical manner.

A. aurita have a much reduced capture surface (shorter tentacles) compared with the Cyanea spp. Heeger & Möller (1987) found that the majority of prey capture by A. aurita in Kiel Harbour, N Germany, occurred on the tentacles as opposed to the subumbrellar surface, so this reduced capture area may account for the low trophic position and narrowest niche width of this species in the present study.

Although they differ in terms of maximum individual size, the congenerics C. lamarckii and C. capillata have similarities in both nematocyst complement (Ostman & Hydman, 1997; Shostak, 1995) and morphology (Holst & Laakmann, 2013). Previous studies have reported predation of C. capillata on A. aurita medusae, therefore it is possible that the differences observed with A. aurita may be a symptom of intra-guild predation by the larger C. capillata (e.g., Hansson, 1997; Purcell, 2003; Titelman et al., 2007).

The isotopic variation found in this study suggests niche partitioning and represents a host of differences in morphology, bell pulsation strength, prey capture techniques and nematocyst composition that enable differential prey capture (Bayha & Dawson, 2010; Costello & Colin, 1994; Peach & Pitt, 2005). Therefore, caution must clearly be taken to avoid over-simplification of jellyfish in ecosystem models. In a broader context, as gelatinous zooplankton span >2,000 species (Condon et al., 2012), occupying habitats ranging from the deep ocean through to shallow water near-shore environments, the inclusion of an ‘average’ jellyfish in such models is likely to underestimate the collective impact in terms of energy flow or consumption of prey (Pauly et al., 2009).

Intra-specific differences in trophic ecology

A. aurita and C. capillata shifted their use of both energy source (δ13C) and trophic position (δ15N) with increasing body size, independent of time (Fig. 4). This suggests that different sized jellyfish medusae, present in the water column at the same time and with access to the same prey field, feed at different positions in the food web (Fleming et al., 2011; Graham & Kroutil, 2001). The simultaneous presence of different sized medusae appears to be a consistent trait across a range of species at temperate latitudes (Houghton et al., 2007), suggesting that jellyfish reproductive cohorts are often poorly defined with a marked overlap within given seasons. C. lamarckii, however, did not exhibit a size-based shift in trophic position with increasing body size. This most likely reflects the comparatively narrow size range of the medusae sampled (3.5–20 cm), with the species rarely exceeding a bell diameter of 30 cm (Russell, 1970). By comparison, C. capillata medusae spanned a far broader size range (6–85 cm) allowing size related shifts in diet to be more easily identified. There are also size related differences in toxicity; although C. lamarckii is as venomous as C. capillata (Helmholz et al., 2007), as both species increase in size, so too do the size of their nematocysts (Ostman & Hydman, 1997). These findings suggest that body size in jellyfish may, to some extent, underpin their capacity to feed at multiple trophic levels through ontogeny. There are some clear exceptions to this rule e.g., small gelatinous species (<12 cm bell diameter) such as box jellyfish Chironex fleckeri and Carukia barnesi have extraordinarily powerful stings that enable them to capture relatively large prey such as larval and small fishes (Carrette, Alderslade & Seymour, 2002; Kintner, Seymour & Edwards, 2005; Underwood & Seymour, 2007).

The trophic position of the jellyfish community over time

When considered as a whole, the δ15N values of the scyphozoan jellyfish community in Strangford Lough increased as the season progressed (Fig. 2), even though baseline levels remained constant. This increase in δ15N was unlikely to be a result of a general increase in size of jellyfish over time, as a range of sizes of each species were collected and analysed each month (see Appendix S1). Given that δ15N baselines were constant across the study period, this indicates that trophic position increased over time. In terms of isotopic niche width, there was an interesting dissimilarity between the start of the season (May) and the following months (June, July and August), suggesting a shift to a broader dietary niche in the latter months (Fig. 3). This increased resource utilisation is consistent with previous studies that suggested jellyfish dietary niches are extremely broad, with species operating as generalists (Dawson & Martin, 2001; Ishii & Båmstedt, 1998; Schneider & Behrends, 1998) feeding opportunistically across a range of plankton (Båmstedt, Ishii & Martlnussen, 1997; Titelman et al., 2007). Therefore, our data suggest that a different and possibly constrained resource pool is being exploited at the beginning of the ‘jellyfish season.’ There are of course environmental factors such as temperature which could have an effect on N metabolism & excretion in jellyfish (Morand, Carre & Biggs, 1987; Nemazie, Purcell & Glibert, 1993) and temperature can have a significant effect on isotopic turnover times in a range of taxa (see Thomas & Crowther, 2015). The temperature increase in Strangford Lough over the course of the study was modest (from 8.7–14.2 °C) but cannot be discounted as a possible influence on isotopic variation over time. The sequential change in species composition seen in Strangford Lough could, in part, be the result of intra-guild predation (Bayha et al., 2012; Robison, 2004; Titelman et al., 2007), which may also contribute to the observed broadening in isotopic niche. Additionally, the collective increase in trophic position over time may reflect species succession in the lough with a general shift from an A. aurita dominated in system in May through to a C. capillata dominated system in August (Fleming et al., 2014). Most likely our results reflect interplay of these two scenarios but highlight the problems associated with assuming that different jellyfish species occupy a single trophic position or ecological niche (Boero et al., 2008; Pauly et al., 2009).

Interspecific and temporal variation in consumer isotopes values can be put into deeper ecological context through the use of models to estimate trophic position (Post, 2002) and consumption patterns (Phillips et al., 2014). However, the use of these models requires reliable estimates of trophic enrichment factors. We welcome the recent TEF estimates made by D’Ambra, Carmichael & Graham (2014) for Aurelia sp.; however, we found that the use of their TEFS resulted in unfeasibly high trophic positions for the Aurelia and other jellyfish in our system. For example, using Post’s (2002) basic model for tropic position resulted in a mean jellyfish trophic position of 17, with the baseline provided by our mean Mytilus δ15N values. As such, realistic estimates of jellyfish trophic level and consumption made using tools requiring accurate TEFS (e.g., mixing models) remain problematic. We therefore call for more experimental work to characterise jellyfish TEFs.

Conclusions

All species showed temporal shifts in their location in δ15N–δ13C space across the study. Given the lack of marked changes at the base of the food web, this suggests that the three jellyfish species consumed different prey across the study period. Size-based shifts in δ13C and δ15N values were evident in two of the three jellyfish species examined here, leading to an inference that variation in body size in some way drives variation in the trophic ecology of a particular species. Distinct differences in δ13C and δ15N values were found within and between species, with evidence of niche segregation between A. aurita and the two Cyanea species. Niche width for all species combined increased considerably throughout the season, reflecting interplay of possible intra-guild predation, temporal shifts in δ13C and δ15N values and the seasonal succession in gelatinous species.

Taken together, these lines of evidence reinforce the idea that scyphozoan jellyfish require more elegant inclusion in ecosystem or fisheries-based models. The salient point here is that jellyfish should not be averaged or defined as a single amorphous group with little reference to temporal and allometric shifts in individual species or gelatinous communities alike.

Supplemental Information

Appendix S1 Appendix S1

Fleming et al raw data (δ13C, δ15N, C:N, mass and bell diameter).

Click here for additional data file.

We are grateful to Phillip Johnston from Queen’s University, Belfast Marine Laboratory (QML) for boat support and sample collection. Thanks also go to Eoin Bleakney, Claire Armstrong, Debbie Baird-Bower and Natalie McCullagh for help with sample processing.

Additional Information and Declarations

Competing Interests

Author Contributions

The authors declare there are no competing interests.

Nicholas E.C. Fleming and Chris Harrod conceived and designed the experiments, performed the experiments, analyzed the data, contributed reagents/materials/analysis tools, wrote the paper, prepared figures and/or tables, reviewed drafts of the paper.

Jason Newton performed the experiments, contributed reagents/materials/analysis tools, wrote the paper, reviewed drafts of the paper.

Jonathan D.R. Houghton conceived and designed the experiments, performed the experiments, analyzed the data, contributed reagents/materials/analysis tools, wrote the paper, reviewed drafts of the paper.

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
