# Peer review of "Not all jellyfish are equal: isotopic evidence for inter- and intraspecific variation in jellyfish trophic ecology"

_PeerJ, doi:10.7717/peerj.1110_

## Round 0.1 · original submission · Minor Revisions

All three reviews were positive and I agree with them, so please respond to their suggestions to improve the paper.

Reviewer 1 ·

Basic reporting

No comments

Experimental design

No comments

Validity of the findings

No comments

Additional comments

The authors address an important topic. The general approach is robust and I think that the manuscript should be accepted after minor revision. I have one major concern regarding interpretation of baseline signatures that the authors should address carefully. Other comments are generally minor.
Major comment
Paragraph starting line 246 - Baseline d15N values remained constant for the duration of the study but baselines were measured using filter feeding and grazing molluscs as surrogates for directly measuring isotopic signatures of phytoplankton and benthic algae. What is the turnover time of N in molluscs? The study only ran for 4 months and it is possible that the baseline may have shifted but slow turnover times in the molluscs may have prevented this being observed. If the jellies responded more quickly to a shift in the baseline than the molluscs then the pattern could still have reflected a change in the baseline. I think it would be useful to provide information on turnover times of N in molluscs and jellies to provide a more robust argument for the pattern being caused by a change in trophic status of the jellies.

Minor comments
Abstract – how was the value of 1400 species of gelatinous zooplankton derived? I believe that there are more than 1000 species of hydromedusae alone.
Line 31 – size-based shifts in what? Presumably diet? Note that this sentence actually doesn’t make sense, so please rewrite.
Line 174 – delete the first sentence as there is no need to repeat information already provided in the methods.
I found the structure of the discussion a little convoluted. The authors first discuss the results of two-way ANOVAS comparing species and time (for months of June/July only) for isotopic centroids (Fig 2) and SEAB(Fig 3) They then discuss size/ontogenetic changes in isotopic centroids (Fig 4) and then switch back to comparing temporal variability in isotopic centroids and SEAB for individual species (thereby referring back to Figs 2 & 3 again). I think the results would flow better if they were rearranged such that temporal variability in centroids (both the two-way comparison between June/July and for individual species) were discussed first, then temporal variation in SEAB discussed second, and size differences in centroids discussed third. If they do this then the reader will be able to progress through each of the figures in a logical manner, instead of switching back and forth between them.
Paragraph starting line 254 - It would be useful to see how the temporal changes in isotopic centroids/isotopic niche width reflected size/ontogenetic differences through time.
Fig 2 – it would be useful to indicate on the figures which species and months differed so that the reader doesn’t need to keep referring to the text to interpret patterns.
Lines 273 – I found the first paragraph of the discussion quite odd and, for the most part, irrelevant. Why is it necessary to discuss the value we assign to jellies, potential expansion of jelly populations or the potential causes of jellyfish blooms? This information doesn’t really seem relevant to the topic of trophic diversity of jellies (sorry but I’m really tired of reading these types of statements – let’s keep to the topic!!). I recommend starting the discussion at line 283.
Line 290 – “inferring” is used incorrectly here.
Line 302 – please provide a reference to support the claim that the cnidomes vary among individuals.
Line 302 – 305 – how does a reduced cnidome and reduced capture surface relate to Aurelia having a low trophic position? Further discussion of how the cnidome of Aurelia relates to the types of prey it captures is required.
Line 307 – “…yet differed here with regard to their del13C and del15N values and niche widths.” but the centroids and isotopic niche width of both Cyanea species overlapped during June and July (see lines 187-188 and 195-197) so this interpretation seems only partially supported by the data. My feeling is that the authors need to provide a more nuanced discussion about differences in isotopic signatures and niche widths given that differences among species were observed among some times but not others.
Line 376 – “in” missing between “variation” and “body” also “dives” should be “drives”

Reviewer 2 ·

Basic reporting

Introduction:
-You often mention the importance of elaborating models that include the jellyfish as one functional group, and in lines 38-41 mention that 100 models were considered, only 23% of which included jellyfish as a distinct functional group. Did any of these focus on your area of study? If so, how might your findings affect the overall model outcome? If better parameterization of these models is an argument for the significance of the study and your study is focused on a specific site, then the link needs to be better expressed.

-How widespread are the three jellyfish species sampled? Have any other isotope values for these or similar species been reported? Is there any background data on what their diets consist of specifically?

Discussion:

-As mentioned regarding the introduction, how applicable are these results to a more broad understanding of jellyfish trophic roles?

-Can you mention any models that might specifically benefit from these findings and how?

Experimental design

Methods:

-Lines 85-89: why were these two species alone chosen to assess changes in baseline 15N? Additionally, you assess changes in isotopic niche space over time and attribute these changes to variation in food resources. Why then are baseline 13C values not reported to account for the changes within the same potential food resources?

Results:

-Figure 2 seems to suggest that all species considered had a shift in isotopic signature between May and June in both 13C and 15N – do you infer a reason for this?

Validity of the findings

No comment

Additional comments

Overall, I thought this paper did a good job describing inferred changes in jellyfish diet over a few month period at a specific site. The statistical methods seemed robust and suitable for the expressed difficulties obtaining even sample sizes at all times considered. It did support the authors’ point that their trophic role should be considered more specifically than an aggregate functional role, but I thought that significance could be explained better.

·

Basic reporting

No Comments

Experimental design

No Comments

Validity of the findings

No Comments

Additional comments

I have really enjoyed reading the Fleming et al. manuscript in which the authors investigated the trophic ecology of three co-occurring jellyfish. Stable isotopes (δ13C and δ15N) were used to address the following three points: 1) to investigate the inter-specific differences in trophic ecology, 2) to detect the possible size-base shifts in trophic ecology and 3) whether the jellyfish tropic position and isotopic niche remain constant over time. In accordance to the results they concluded that the three selected species occupy different trophic position. Further, the isotopic niche width combined for all three species was not constant over the investigated period; in fact, the isotopic niche width increased throughout the season, reflecting temporal shifts in trophic position and seasonal succession in the investigated species.
The manuscript is very well structured, the figures and tables being very informative. I particularly like the detailed description of the methodology and data analysis. The relevant results to address the initial hypotheses are described and very well discussed. However, there are few comments I would like the authors to address, in particular I am concerned about the methodology used for drying the samples and how this might have affected the stable isotope composition of gelatinous tissues. Once the authors discuss the above mentioned issue, I believe the manuscript will be acceptable for publishing.
Specific comments:
Line 42:
The reference Graham & Kroutil, 2001 report on prey types only for A. aurita and intra-specific differences are not mentioned. Please, remove or refer to it when writing about the ontogenetic shifts in jellyfish diet.
Line 51:
I think a better reference would be Purcell 2009 instead of Purcell 1992. (Please see below for the complete reference)
Material & Methods
As already mentioned above, I am concerned that the stable isotope composition of the samples might have been affected during the process of sample preparation. The samples were oven dried at 60°C. This is a standardised drying method commonly used for drying zooplankton. However, a recent work (even cited by the authors (Kogovšek et al. 2014)) revealed that exposing gelatinous tissues to 60°C may affect stable isotope composition of the jellyfish. This change was not uniform among the species; it seems that the change in δ15N was more pronounced in the species that are more protein rich compared to the protein depleted one. In the current study, the protein rich Cyanea tissue may therefore be more affected when exposed to the drying temperature than Aurelia aurita and thus affecting the δ15N composition in a non equal manner. I would like the authors to comment this in the discussion section of the manuscript.
Discussion
(line 314, 315) From the current text it is not clear whether the differences in δ15N are due to direct effect (differences in size of nematocysts and toxins may result in different quality and quantity of proteins) or indirect, as the different pray captured and consumed.
Some previous studies report on predation of Cyanea on Aurelia medusae (for example Hansson 1997, Purcell 2003). I miss this point in the discussion when addressing inter-specific interactions.
Further, the turnover rates of the isotopes in the jellyfish tissues are unknown. However, we can assume that the turnover rates (bulk or tissue specific) would be affected by the temperature and/or the food quality, as for example was previously observed in fish. In the current study, the samples from May were significantly different from other months for all species. May this be in part a consequence of the lower temperature in May compared to other months? Or may be the spring zooplankton bloom? I think it would be useful to include this information in the discussion too.
There are some small typing mistakes throughout the text.
References:
Hansson, L. J. (1997). "Capture and digestion of the scyphozoan jellyfish Aurelia aurita by Cyanea capillata and prey response to predator contact." Journal of Plankton Research 19(2): 195-208.
Purcell, J. E. (2003). "Predation on zooplankton by large jellyfish, Aurelia labiata, Cyanea capillata and Aequorea aequorea, in Prince William Sound, Alaska." Marine Ecology Progress Series 246: 137-152.
Purcell, J. E. (2009). "Extension of methods for jellyfish and ctenophore trophic ecology to large-scale research." Hydrobiologia 616(1): 23-50.

---

## Round 0.2 · accepted · Accept

I think you did a good job of responding to all reviewer comments.